# Spatial-temporal Graph Attention Network for Forex Rate Forecasting with Hierarchical Transformer

## Abstract

The forex market, with its daily trading volume reaching nearly trillions of dollars, presents significant opportunities for the application of advanced predictive analytics. Traditional forex forecasting methods often overlook the interdependencies between currencies and struggle with long-range data dependencies, leading to challenges in capturing the true market dynamics. To overcome these limitations, we propose a novel Spatial-Temporal Graph Attention Network with Hierarchical Transformer (STGAT). Our model innovatively combines spatial graph convolutions with a dual-view temporal transformer-based mechanism, utilizing a temporal linearity graph attention network to account for currency relations in a time-sensitive manner. By integrating a linear attention mechanism for enhanced efficiency and capturing both local and global sequential data embeddings, STGAT provides a framework based on a hierarchical transformer for predicting exchange rates. We validate our approach on exchange rates of seventeen currencies over 2,092 trading days, demonstrating superior performance compared to state-of-the-art models.

## 1 Introduction

The daily trading volume of the forex market reached nearly trillions of dollars Islam & Hossain (2021). The exchange rates of various currencies provide a large amount of important data for the currency exchange market. The forex rate (FR) not only affects the profits of enterprises and financial institutions, but also influences the development of the international trade market and the adjustment of national monetary policy Wang et al. (2021). Therefore, FR prediction is crucial to managing financial risk, making informed investment decisions, planning economic policies, and ensuring competitive international pricing and profitability Pradeepkumar & Ravi (2018); Jarusek et al. (2022).

Traditional solutions for forex prediction are based on time-series analysis models. Recent advances in deep learning present a promising prospect in the prediction of forex rates using long-short-term memory (LSTM)-based methods Datta et al. (2021); Islam & Hossain (2021); Dautel et al. (2020). In addition, some work Islam & Hossain (2021); Dautel et al. (2020) investigates the performance of the gated recurrent unit and LSTM models in predicting exchange rates. Following it, Bi-LSTM Datta et al. (2021) is applied to forecast the exchange rates of 22 different currencies against the US dollar. Moreover, the paper Mao et al. (2024) studies the performance of a hybrid model combining a convolutional neural network with transformer Vaswani (2017) and CNN-LSTM models for exchange rate prediction.

However, such the above works for FR prediction have two challenges. First, they treat FR prediction as independent of each other, which is contrary to true market function, and makes forecasting challenging. FR is interconnected due to economic policies and trade relationships Hartmann (1998); Auboin & Ruta (2013); Dell'Ariccia (1999). For example, if the U.S. Federal Reserve raises interest rates, it can strengthen the USD, which in turn impacts other currencies pegged to the USD or those in countries with close economic ties to the United States. Figure 1 shows that the currencies GBP, ERU and CHF share highly coincident price movements, and the same scenario can be found for the currencies HKD and MOP. In addition, for trading, if two countries trade extensively, changes in the

exchange rate of one currency can directly affect the trade balance, influencing the other currency. For the second challenge, existing work cannot handle long-range dependencies within the data and the importance of different parts of the input data, regardless of their position. Therefore, it can suffer from vanishing gradients when dealing with long input sequences.

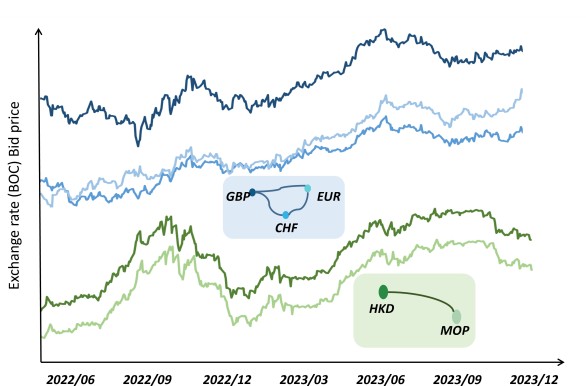

Figure 1: Exchange currency ratios GBP, EUR and CHF show correlated trends and share highly coincident price movements

To address the above two challenges, we propose a **S**patial-**T**emporal **G**raph **A**ttention network with hierarchical **T**ransformer (STGAT). STGAT learns the collective correlation between currencies by combining spatial graph convolutions with a dual-view temporal transformer-based mechanism through graph attention to capture the spatial and temporal dependencies in FR. Specifically, we first feed the historical time series data of FR to a CNN and then process them by a hierarchical transformer for both local and global sequential embedding. By devising a new temporal linearity graph attention network (TLGAT), we next revise the sequential embeddings by considering currency relations in a time-awareness way to generate the rational embedding while improving efficiency. Finally, we adapt the relational embeddings to a fully connected layer to predict the FR rate. We used seventeen currencies and show the superior performance of STGAT over the state-of-the-art models. The contributions of our work can be summarized as:

- We propose a novel spatio-temporal graph attention network with a hierarchical transformer (STGAT) for FR prediction.
- We combine linear attention mechanism with spatial graph convolutions through linear attention to improve efficiency.
- Through experiments on seventeen currencies over 2,092 trading days, we demonstrate the superior performance of STGAT over the state-of-the-art methods.

## 2 RELATED WORKS

### 2.1 FOREX PREDICTION WITH TIME SERIES MODELING

In recent years, numerous machine learning techniques have been applied to the field of exchange rate forecasting. Among them, neural network-based modeling has been shown to be effective in predicting exchange rate movements (Datta et al., 2021). In 2020 and 2021, Islam & Hossain (2021), as well as Dautel et al. (2020), investigate the performance of the GRU and LSTM models to predict FR. Their findings indicated that the combined GRU-LSTM model performs exceptionally well. In 2021, Datta et al. (2021) applies a Bi-LSTM model to forecast the exchange rates of 22 different currencies against the US dollar. Furthermore, in 2024, Mao et al. (2024) explores the performance of the CNN transformer and CNN-LSTM models for the prediction of exchange rates, the results indicating that CNN-LSTM outperformed other models in most cases. Similarly, in 2024, Vibhute et al. (2024) examines the performance of three variants of LSTM in exchange rate forecasting, which concluded that bidirectional LSTM demonstrated the best results. However, the above methods do not directly take into account the correlation between different currencies, and modeling dependency with a graph structure is widely used Ge et al. (2021).

### 2.2 SPATIAL-TEMPORAL GRAPH LEARNING IN FINANCE

There are some studies of graph attention networks (GTAs) in other time series forecasting areas Qiao et al. (2023); Zhang et al. (2020). The paper Zhong et al. (2023) successfully predicted the

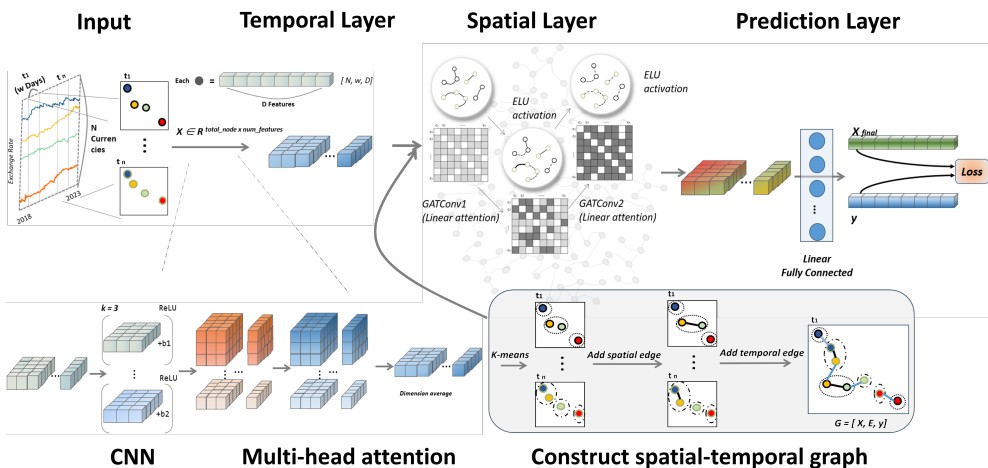

Figure 2: Framework of Spatial-Temporal Graph Attention network with hierarchical Transformer (STGAT). STGAT contains three layers that are the temporal layer, spatial layer and prediction layer that outputs the next day's FR.

trend in the price of cryptocurrencies such as Bitcoin with a combination of the LSTM and Re-GAT model based on the GAT model. In this research, LSTM is used to capture the temporal trend of cryptocurrencies and ReGAT is used to capture the correlation between different types of cryptocurrencies. Han et al. (2024) uses the LSTM-GAT model to predict multivariate time series data. Firstly, they capture the long-term dependence of features in time-series data through LSTM, afterwards they convert the features to graph-structured data, and finally they use GAT to capture spatial features in multivariate time-series data. The paper Zhao et al. (2024) used a hybrid model with Informer and GAT for time series anomaly detection, where Informer is used to capture long-term dependencies in time series and GAT is used to capture correlations between multivariate time series. The results indicate that all of these models have good performance behavior. Wen et al. (2022) showed that combining the graph neural network model with the Transformer model could lead to a general better understanding of temporal and spatial correlation and significantly improve the model performance. Wen et al. (2022) also stated that how to use these two models for spatial-temporal modeling of time series data is an important future research direction.

At present, there is limited research using GAT for its forecasting in the exchange rate time series forecasting domain. This indicates that our model is innovative and academically significant in this field. It effectively fills the gap in the literature in this field. Above all, these studies have shown that GAT can effectively capture spatial correlation among multivariate time series data. This will provide strong theoretical support for our model.

## 2.3 METHODOLOGY

### 2.3.1 PROBLEM FORMULATION

We formulate the prediction of FR as a learning-to-regression problem. Let $C = \{c_1, c_2, \ldots, c_N\}$ denotes the set of $N$ currency pairs, where for each currency pair $c_i \in C$ on trading day $t$, there is an associated exchange rate $p_i^t$. On any given trading day $t$, we aim to learn a function $f$ such that $\hat{p}_i^{t+1} = f(c_i, t)$, where $\hat{p}_i^{t+1}$ is the predicted exchange rate for currency pair $c_i$ on day $t + 1$.

We demostrate our framework in Figure 2. In the following subsections, we first show how the convolutional operation is used to extract local feature, and explain the hierarchical transformer for globally learning the temporal evolution of currency features (Section 2.3.2). We then describe the construction of the forex temporal graph with $k$-means (Section 2.3.3). Finally, we summarize the temporal and spatial graph layer and optimize the framework to capture temporal and spatial dependencies for end-to-end forex prediction.

### 2.3.2 Hierarchical Transformer with Convolutional Operation

In recent years, the transformer model shows great potential to handle time series data forecasting (Vaswani, 2017; Liu et al., 2023) due to the ability to model long dependencies and dynamic attention weights. However, we still face two challenges that are lack of local time-series information and high computational complexity. To mitigate them, in this paper, we use convolution operation to enhance local time-series information and an effective linear attention mechanism to reduce computational complexity.

**Convolutional Mechanism.** This mechanism directly uses one-dimensional convolutional operations to capture local features and maintain temporal continuity. The input to a convolutional layer is a time-series forex ratio data. Let $x$ be a one-dimensional input sequence of length $L$, represented as: $x = [x_1, x_2, \ldots, x_L]$. The convolution applies filters to the input data to produce feature maps. Suppose we use a filter $f$ of size $K$, the convolution operation at a position $t$ in the input sequence is computed as:

$$y_t = \sum_{k=0}^{K-1} f_k \cdot x_{t+k} + b, \tag{1}$$

where $y_t \in \mathbf{Y}$ is the output at position $t$, $f_k$ is the value of the filter at position $k$, $x_{t+k}$ is the input at position $t + k_1$, $b$ is a learnable bias term, $K$ is the size of the filter. The filter is slid over the input data, typically with a certain stride $S$ to produce a sequence of outputs $y$, which form the feature map. The stride $S$ determines the step size of the filter moving over the input.

**Hierarchical Transformer.** After completing the covolutional operation, we need to capture the global temporal dependence of the features through positional encoding and multi-head attention mechanisms of transformer. We first perform the position encoding (PE) that is used to add unique positional information to the input features to ensure that the multi-head attention mechanism can perceive the time-step order of the input sequence. Generate a specific position vector for each dimension of the characteristic at each time step $t$ using sine and cosine functions.

$$\text{PE}(t, 2i) = \sin\left(\frac{t}{10000^{\frac{2i}{d}}}\right) \tag{2}$$

$$\text{PE}(t, 2i+1) = \cos\left(\frac{t}{10000^{\frac{2i}{d}}}\right) \tag{3}$$

where $d$ represents the feature dimension of the convolutional layer output, $t$ denotes the position index of the current time step; $2i$ and $2i + 1$ represent the even and odd indexes of the feature dimensions with a range from $0$ to $d$. After each feature is computed for each time step, the positional encoding is stored as a matrix. Next, the PE is summed element-by-element with the input features to generate an up-to-date feature containing both local and positional information. Finally, the features tensor $\mathbf{X} = \mathbf{Y} + \text{PE}$ are taken as input to the multi-head attention.

For multi-head attention, we divide the input features into multiple subspaces, and in each subspace, the attention weights are computed and aggregated in parallel. Firstly, the input features $\mathbf{X}$ are mapped to query, key, and value.

$$\mathbf{Q} = \mathbf{X}\mathbf{W}_Q, \quad \mathbf{K} = \mathbf{X}\mathbf{W}_K, \quad \mathbf{V} = \mathbf{X}\mathbf{W}_V, \tag{4}$$

where $\mathbf{Q}$ is used to help the model find which time steps have the highest correlation with the current time step; $\mathbf{K}$ is used to determine whether the time steps are related to each other; $\mathbf{V}$ is used to represent the actual feature information at each time step; $\mathbf{W}_Q, \mathbf{W}_K, \mathbf{W}_V$ are linear transformation matrices. Afterwards, the correlations existing between different time steps are computed globally through the attention mechanism to generate feature representations containing global temporal dependencies. For the encoder head$_i$,

$$\text{head}_i = \text{Attention}(\mathbf{Q}\mathbf{W}_i^Q, \mathbf{K}\mathbf{W}_i^K, \mathbf{V}\mathbf{W}_i^V) = \text{softmax}\left(\frac{\mathbf{Q}\mathbf{W}_i^Q \cdot (\mathbf{K}\mathbf{W}_i^K)^T}{\sqrt{d_k}}\right)\mathbf{V}\mathbf{W}_i^V, \tag{5}$$

where $\mathbf{Q}\mathbf{W}_i^Q$, $\mathbf{K}\mathbf{W}_i^K$, $\mathbf{V}\mathbf{W}_i^V$ denote the linear transformation of the original $\mathbf{Q}$, $\mathbf{K}$, $\mathbf{V}$ to obtain the $\mathbf{Q}$, $\mathbf{K}$, $\mathbf{V}$ vector corresponding to the current head; $d_k$ is the dimension of Key, which is used to

prevent the inner product from being too large. We then splice the output of all heads:

$$\text{MultiHead}(\mathbf{Q}, \mathbf{K}, \mathbf{V}) = \text{Concat}(\text{head}_1, \text{head}_2, \dots, \text{head}_h)\mathbf{W}^O, \tag{6}$$

where $\mathbf{W}^O$ is the matrix used to linearly transform the final result. The aim is to map the feature dimensions to the initial feature dimensions.

As of now, the feature tensor after processing by the temporal information module is denoted as $\mathbf{X}$. In order to further aggregate the local and global features as well as to ensure that the subsequent graph attention network layer can efficiently process the spatial information in conjunction with edge indexing, we perform feature dimension averaging on the feature tensor $\mathbf{X}$. After averaging, the features are represented as $\mathbf{X}'$. This step can be represented as: $\mathbf{X}'_{i,j} = \frac{1}{d}\sum_{k=1}^{d}\mathbf{X}_{i,j,k}$. Finally, to mitigate overfitting in subsequent modules, we also use Dropout regularisation technique to discard some entities of features randomly.

### 2.3.3 TEMPORAL GRAPH ATTENTION MECHANISM

**Temporal Forex Graph Construction with $k$-means.** We model currency interdependence via graph structure, where nodes represent currencies and edges represent relations between currencies. We suppose that currencies within the same cluster at each time step should have similar behavioral patterns at time step $t$, and hence constitute the spatial relationships represented by edges in our spatial-temporal graph. In addition, $k$-means is widely used to reveal temporal and spatial patterns in some applications, such as power grid systems Gharavi & Hu (2017) and traffic flow analysis Tian et al. (2024). Inspired by them, we construct this spatial-temporal currency graph using the $k$-means algorithm.

To create edges within a cluster, we suppose $x_{i,t}$ is the feature vectors of node$_i$ at time step $t$, we generate a matrix of cluster labels such as $L_t = [l_{1,t}, l_{2,t}, \dots, l_{n,t}]$ and $l_{i,t} \in \{0, 1, \dots, k-1\}$ are the cluster labels of node$_i$ at time step $t$. we create edges within clusters. Firstly, we need to obtain the index of the node that belongs to the same cluster $C_j$: $\text{indices}_i^t = \{i \mid l_i^t = j\}$, where $\text{indices}_i^t$ denotes the set of all node indices labeled $j$ within the current time step $t$, which can also be referred to as the set of local node indices. Firstly, we have the index of the node belonging to the same cluster $C_j$: $\text{indices}_i^t = \{i \mid l_i^t = j\}$. Where $\text{indices}_i^t$ denotes the set of all node indices labelled $j$ within the current time step $t$, which can also be referred to as the set of local node indices. To ensure the existence of unique identifiers for nodes at different time steps in the final generated spatial-temporal graph, we combine the time step $t$ with the local index $i$ of a node to generate the global index $\text{global\_index}_i^t = i + t \times n$, where $i$ is the local index of a node at time step $t$; $n$ is the number of nodes in each time step; $t \times n$ is the offset of the time step. Finally, we create the undirected fully connected edge within the cluster and update the edge index.

To create edges across clusters, in addition to representing the spatial correlation between different nodes, we need to reveal the temporal correlation of the same node over the complete time period. We connect the feature representations of the same node at neighboring time steps. Such one-way edges ensure that the direction of the edge points from time step $t$ to time step $t + 1$, which in turn ensures that edges across time steps can represent temporal dependencies that are consistent with the temporal order. Different from the previous section, this section treats the feature representation of the same node (currency) in neighboring time steps as two separate nodes, which in turn generates edge indexes across time steps.

**Graph Attention.** Graph Attention Networks (GATs) Veličković et al. (2017) are a type of neural network architecture designed specifically for graph-structured data. It utilizes the attention mechanism to weigh the importance of nodes' features during information aggregation. This allows GATs to dynamically prioritize information from different neighbors, enhancing their capability to capture node interdependencies without the need for costly matrix operations.

The GAT needs to combine the features of the source and target nodes to calculate the attention coefficient, we first need to vary the input node features linearly and get the source node feature $\mathbf{X}_{\text{src}}$ and target node feature $\mathbf{X}_{\text{dst}}$:

$$\mathbf{X}'^{(h)}_{\text{src}} = \mathbf{W}^{(h)}\mathbf{X}_{\text{src}}, \quad \mathbf{X}'^{(h)}_{\text{dst}} = \mathbf{W}^{(h)}\mathbf{X}_{\text{dst}} \tag{7}$$

where $\mathbf{W}^{(h)}$ is the linear transformation matrix; $\mathbf{X}'$ is the linearly transformed matrix. $\mathbf{X}'_{\text{src}}$ and $\mathbf{X}'_{\text{dst}}$ are the input features of the source and target nodes; $h$ represents the $h$-th attention head.

Subsequently, the attention coefficient in GATs quantifies the influence of neighboring nodes, but as the network scales, the computational complexity increases significantly, as demonstrated by some work Yang et al. (2024); Shen et al. (2021). In response, the paper Katharopoulos et al. (2020) proposes to simplify nonlinear attention to a linear form, offering a potential solution to manage computational costs and inspire further research on attention mechanisms. Motivated by them, we simplify the computation of the attention mechanism of GAT to a linear version that directly uses a linear combination of node features to compute the attention weights. The calculation of the attention coefficient can be expressed as:

$$\alpha_{\text{src,dst}}^{(h)} = \text{softmax}\left(\text{att}_{\text{src}}^{(h)} \cdot \mathbf{X}'^{(h)}_{\text{src}} + \text{att}_{\text{dst}}^{(h)} \cdot \mathbf{X}'^{(h)}_{\text{dst}}\right) \tag{8}$$

where $\text{att}_{\text{src}}^{(h)}$ and $\text{att}_{\text{dst}}^{(h)}$ represent the attention weight vectors of the source and target nodes respectively; $\mathbf{X}'_{\text{src}}^{(h)}$ and $\mathbf{X}'_{\text{dst}}^{(h)}$ are the source node features and target node features after linear transformation; $\alpha_{\text{src,dst}}^{(h)}$ denotes the attention coefficient between the source node and the target node in the $h-$th attention head; softmax$(\cdot)$ denotes the normalisation operation.

After computing the attention coefficients, we need to generate a new feature representation by weighted aggregation the features of the source node to the target node through these attention coefficients. So, we have:

$$h_{\text{dst}}^{(h)} = \sum_{\text{src} \in \mathcal{N}(\text{dst})} \alpha_{\text{src,dst}}^{(h)} \cdot \mathbf{X}_{\text{src}}^{(h)} \tag{9}$$

where $h_{\text{dst}}^{(h)}$ is the feature representation generated by the target node at the $h$-th attention head; $\mathcal{N}(\text{dst})$ denotes the set of all neighbouring nodes of the target node; $\alpha_{\text{src,dst}}^{(h)}$ is the normalised attention weight computed by the neighbour node on the target node; $X_{\text{src}}^{(h)}$ is the feature representation of the source node. In order to compensate for the lack of information caused by the linear attention mechanism, we perform a nonlinear transformation of the aggregated features by introducing a nonlinear activation function.

$$h_{\text{dst}}^{(h)} = \text{ELU}(h_{\text{dst}}^{(h)}) \tag{10}$$

where the ELU is calculated by this formula:

$$\text{ELU}(z) = \begin{cases} z & \text{if } z > 0, \\ \alpha(\exp(z) - 1) & \text{if } z \leq 0 \end{cases} \tag{11}$$

$\alpha$ is a hyper-parameter.

In the end, we will get a node feature $\mathbf{X}_{\text{final}}$ that contains both temporal and spatial information for each currency. Converting the features which contain all the spatial-temporal information into a final concrete output value by means of a linear layer. This step can be represented as:

$$\mathbf{Z} = \mathbf{X}_{\text{final}}\mathbf{W}_{\text{out}} + \mathbf{b}_{\text{out}} \tag{12}$$

where $\mathbf{W}_{\text{out}} \in \mathbb{R}^{\text{hidden\_channels} \times \text{out\_channels}}$ is a learnable matrix, $\mathbf{b}_{\text{out}}$ is the bias term of the linear layer; $\mathbf{Z}$ is the final output of predicted currency exchange ratio.

## 3 EXPERIMENTS

### 3.1 EXPERIMENTAL SETTTINGS.

**Datasets.** We use the exchange rate prices of 17 currencies against the Chinese Yuan from the Sina Finance FX market dataset [1]. In this paper, we use the Bank of China Exchange Bid price that represents the price at which the user actually buys the exchange rate through a cashless transaction. The price data of 17 currency exchange rates from 01/January/2018 to 31/December/2023. In addition, we standardize all the initial data, which in turn stabilizes and enhances the model learning performance. We will release our codes alongside datasets through the Github.

**Traing setup.** We perform all experiments on GeForce RTX 4090, 24G Memory and Windows OS. We use window size 100, the number of clusters 3, CNN kernel size 3, learning rate 0.005, dropout

---

[1]http://biz.finance.sina.com.cn/forex/forex.php

Table 1: Performance Comparison. The best results are in **bold** and the second best ones are underlined.

| Category | Model | MAE | RMSE | $R^2$ |
|---|---|---|---|---|
| Regression-based models | XGBRegressor | 1.0388 | 1.2072 | -0.4551 |
| | Lasso | 1.1857 | 1.3998 | -0.9562 |
| | LSTM | 1.1857 | 1.0054 | -0.0092 |
| Transformer-based models | FTTransformer | 0.9418 | 1.0110 | -0.0206 |
| | Flowformer | 0.2166 | 0.3074 | 0.9047 |
| | iTransformer | 0.1937 | 0.2729 | 0.9249 |
| GNN-based models | EdgeConv | 0.9639 | 1.0927 | 0.1757 |
| | GCN | 0.4310 | 0.6551 | 0.7038 |
| | GraphSAGE | 0.5613 | 0.7016 | 0.6602 |
| | FourierGNN | 0.3823 | 0.2370 | 0.7405 |
| | **STGAT** | **0.1377** | **0.2495** | **0.9570** |

rate 0.6, stride and padding size 1. we standardize all data with $X_s = \frac{X - \mu}{\sigma}$, where $X$ represents the original data, $\mu$ represents the mean of the data, $\sigma$ represents the standard deviation of the data, and $X_s$ represents the standardized data.

**Baselines.** We compared our proposed method with ten state-of-the-art models including: 1) regression-based methods: XGBRegressor(Mahmud et al., 2024), Lasso(Tibshirani, 1996), LSTM(Hochreiter, 1997); 2) transformer-based methods: FTTransformer(Gorishniy et al., 2021), Flowformer(Wu et al., 2022), iTransformer(Liu et al., 2023); 3) GNN-based methods: Edge-Conv(Wang et al., 2019), GCN(Kipf & Welling, 2016), GraphSAGE(Hamilton et al., 2017), FourierGNN(Yi et al., 2024).

**Evaluation Metrics.** Mean absolute error (MAE) quantifies the mean of the absolutely difference between the forecast and the actual value. It provides a clear indication of the overall magnitude of the error and is utilized to visually evaluate the performance of model. It is defined as: $\text{MAE} = \frac{1}{n} \sum_{i=1}^{n} |z_i - \hat{z}_i|$.

The root mean square error (RMSE) serves as a standardized measure of the forecast error, evaluating the differences between the predicted values and the actual values. A lower RMSE indicates that the model predictions are closely aligned with real trading data, enhancing the reliability of the model. We formulate it as: $\text{RMSE} = \sqrt{\frac{1}{n} \sum_{i=1}^{n} (z_i - \hat{z}_i)^2}$.

The coefficient of determination ($R^2$) describes the proportion of variability in the response variable explained by the model through the predictor variables. The value $R^2$ indicates the fit of the model to the data and thus the ability of the model to account for the factors that affect the movements of the stock price. It is defined as: $R^2 = 1 - \frac{\sum_{i=1}^{n} (z_i - \hat{z}_i)^2}{\sum_{i=1}^{n} (z_i - \bar{z})^2}$

## 3.2 PERFORMANCE COMPARISON

We compare our STGAT with ten baselines and show the results in Table 1. We observe that: First, our STGAT model consistently has the best performance and the iTransformer is the second best. This is because transformer-based models focus only on capturing correlations in the time dimension, ignoring spatial correlations between different currency exchange rates. Second, overall, transformer-based models and GNN-based models show better performance than performance regression-based models. Thirdly, the regression-based models (e.g., XGBRegressor and Lasso) and even the LSTM (a type of neural network) have notably poor R² values. XGBRegressor and Lasso have negative R² values (-0.4551 and -0.9562, respectively), which suggests that these models perform worse than a simple mean-based prediction model. This could indicate a poor fit of these models to the data or that the data characteristics are not well-suited for linear approaches.

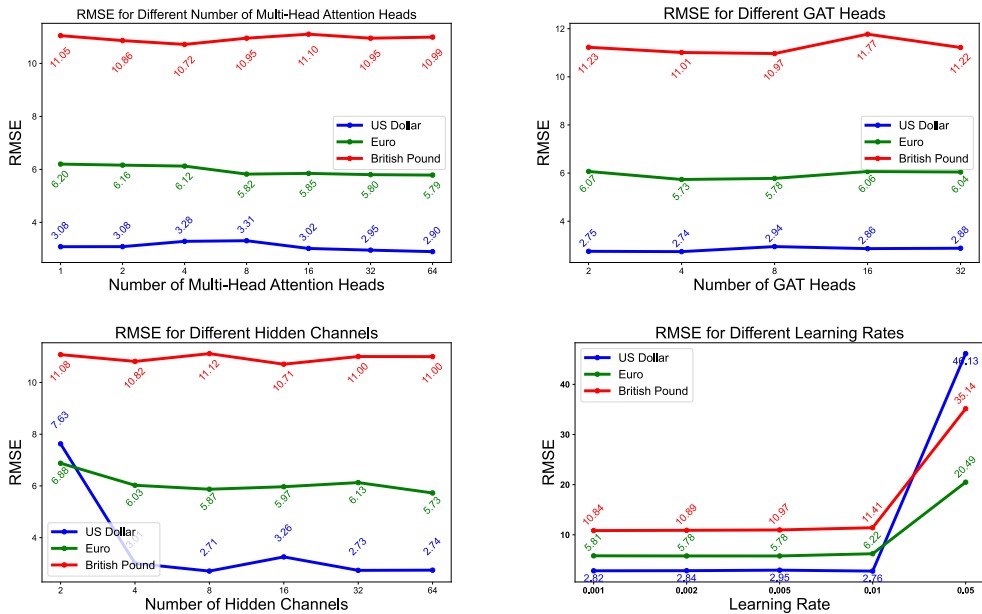

Figure 3: Sensitivity to hyper-parameters: number of multi-head attend heads, number of GAT heads, number of hidden channels and learning rate.

## 3.3 SENSITIVITY ANALYSIS

In order to ensure that our model can demonstrate the best performance with the most appropriate parameters, we carry out a more in-depth and specific analysis of three of the most important hyperparameters. In addition, in order to demonstrate more specific model prediction performance, we have chosen the US dollar, the Euro, and the pound sterling to analyze the performance of our model when dealing with mainstream currencies. The United States dollar play an important role in forex, and it is widely used in international trade and financial markets Chinn et al. (2024). Euro is the second largest reserve currency after the US dollar Chinn et al. (2024). Pound sterling has a high international standing in international markets. However, as a result of the Brexit event, there will be an increase in the volatility of the Pound, especially in the face of uncertain events Zhu (2024). We have the following observations:

- **Head in Multiple Attention Mechanisms:** The model has the best performance when the number of heads in the multi-head attention mechanism is 64. However, we find that the model improves less when the number of heads exceeds 16. This phenomenon suggests that appropriately increasing the number of Attention heads can help the model to capture more complex spatio-temporal features in general, but the enhancement of the number of Attention heads has limited contribution to the overall performance beyond a certain threshold.

- **Heads in GAT Attention Mechanisms:** Further increasing the number of GAT heads does not significantly improve the overall performance, but rather slightly increased the prediction error. This is because a fact that with too many attention heads increase complexity but fails to capture more useful features, leading to overfitting or instability of the model.

- **Hidden Channels:** The model performance increases significantly when the number of hidden channels is increased to 4 or 8, and the model has the best overall performance when the number of hidden channels is 64.

Table 2: Component Ablation Study over STGAT. The best results are in **bold**.

| Model Component | MAE | RMSE | $R^2$ |
|---|---|---|---|
| Transformer + Linear GAT | 0.1590 | 0.2643 | 0.9518 |
| CNN + Transformer + Nonlinear GAT | 0.1388 | **0.2483** | **0.9574** |
| CNN+ Transformer | 0.3062 | 0.5381 | 0.9982 |
| Linear GAT | 0.1533 | 0.2678 | 0.9505 |
| Nonlinear GAT | 0.1546 | 0.2642 | 0.9518 |
| STGAT | **0.1377** | 0.2495 | 0.9570 |

### 3.4 MODEL COMPONENT ABLATION STUDY

In Table 2, our STGAT model with completed components can significantly benefit the improvement of performance. According to the second and third rows, the non-linear GAT can improve the performance. This is because CNN can effectively extract local features. Our model with non-linear GAT has better performance (second row) than our STGAT model with respect to RMSE and $R^2$. However, STGAT has high efficiency because of the usage of linear GAT. According to it, we should make a trade-off between efficiency and effectiveness by considering the linearity.

## 4 CONCLUSION

In this paper, we propose a novel model Spatial-Temporal Graph Attention Network with Hierarchical Transformer (STGAT) for forex rate forecasting that leverages spatial graph convolutions and a dual-view temporal transformer. Our model demonstrates superior accuracy over cutting-edge methods by effectively capturing both spatial and temporal currency dependencies across seventeen currencies and 2,092 trading days. Future work can focus on integrating indicators (e.g., gross domestic product, consumer price index) that reflect macroecnometrics to improve prediction.

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
