# OpenReview forum: "Spatial-temporal Graph Attention Network for Forex Forecasting with Hierarchical Transformer"
_ICLR.cc/2025/Conference — Submitted to ICLR 2025_

### Official Review · Reviewer_SCEZ · 2024-11-01

**Soundness:** 2
**Presentation:** 3
**Contribution:** 2
**Rating:** 5
**Confidence:** 4

**Summary:**

The paper proposes a Spatial-Temporal Graph Attention Network with Hierarchical Transformer (STGAT) for forecasting foreign exchange (forex) rates. This model integrates graph attention networks (GATs) with hierarchical transformers to capture spatial and temporal dependencies across currencies. It introduces a dual-view temporal transformer mechanism to manage both local and global currency correlations, alongside using linear GAT to enhance efficiency. Tested on 17 currencies over 2,092 trading days, the model shows improved performance over state-of-the-art models, including XGBRegressor, LSTM, and FourierGNN.

**Strengths:**

- **Originality**: The combination of spatial graph convolutions with **temporal transformers** is a novel approach to forex rate forecasting, especially as it addresses spatial and long-range dependencies in the data.

- **Quality**: The paper conducts comprehensive experiments, comparing the model against multiple baselines, and presents detailed results using appropriate metrics such as MAE, RMSE, and R².

- **Clarity**: The explanations are clear, particularly in describing the technical contributions like GAT and hierarchical transformer components. Diagrams also effectively aid understanding.

- **Significance**: Forex rate forecasting has significant applications in financial markets, and the accuracy improvement demonstrated in the experiments suggests a practical and meaningful contribution.

**Weaknesses:**

- **Motivation**: The motivation is somewhat weak. While the authors claim that traditional models do not address the interdependencies between currencies and long-range dependencies, other financial time series prediction models, particularly in stock markets, have already tackled similar issues. This diminishes the novelty of the problem formulation.

- **Lack of Macroeconomic Indicators**: The model only uses forex rate data, without considering macroeconomic indicators like GDP or inflation, which are often crucial in driving currency movements.

- **Efficiency vs. Accuracy**: The ablation study suggests that a non-linear GAT could offer slightly better performance than the linear GAT, but the paper does not adequately discuss the trade-off between computational efficiency and potential accuracy improvements.

- **Temporal Granularity**: It is unclear whether the daily data granularity used in the model is optimal. The authors could have experimented with finer temporal granularity, such as hourly or minute-level data, to see if it improves performance.

**Questions:**

- Could incorporating macroeconomic indicators, such as inflation or GDP, improve the model's predictive performance, particularly for long-term predictions?
- Have you explored using finer temporal granularity (e.g., hourly or minute-level data) to capture short-term market fluctuations?
- What are the primary challenges or limitations of applying this model to other types of financial time series, such as stock market prediction?

---

### Official Review · Reviewer_Zvid · 2024-11-01

**Soundness:** 2
**Presentation:** 3
**Contribution:** 1
**Rating:** 1
**Confidence:** 5

**Summary:**

The paper proposes an approach to exchange rate forecasting through the Spatial-Temporal Graph Attention Network with Hierarchical Transformer (STGAT). The model integrates spatial graph convolutions with a dual-view temporal transformer mechanism to capture currency interdependencies and long-range data dependencies. The authors present validation results of STGAT's performance on a dataset covering 2,092 trading days across 17 currencies, indicating some improvement in accuracy over certain existing models.

**Strengths:**

1. This paper focuses on exchange rate prediction, although exchange rate datasets have already become a standard choice in most time-series forecasting models.

2. The authors mention some unique characteristics of exchange rate data compared to other time-series data, though the proposed model does not seem, in my view, to effectively leverage these characteristics.

**Weaknesses:**

1.From a methodological perspective, the paper primarily combines existing graph neural network structures with transformers. The so-called linear attention mechanism is also a common structure, and the approach does not seem to contribute new insights to the ML field.

2.While the authors aim to address exchange rate forecasting, the proposed model feels more like a general time-series forecasting model. Without demonstrating superior performance on commonly used datasets such as ETTh, electricity, or traffic data, it is difficult to find the approach convincing.

3.The experimental results in Table 1 appear quite weak, with many models showing R² values below zero, meaning they perform worse than simple mean prediction. This suggests the authors may not have given adequate attention to data normalization or baseline parameter tuning.

4. The use of k-means for graph construction seems unnecessary, as self-learned graph structures are now common, and many end-to-end deep clustering methods are available.

**Questions:**

Q1: Could the authors test their model on general time-series datasets, given that they compare it with iTransformer?

Q2: The authors should provide more detailed experimental comparisons.

Q3: The effectiveness of using a k-means constructed graph is questionable; could the authors compare it with self-learning graph methods proposed in papers like Graph WaveNet?

---

### Official Review · Reviewer_ucBh · 2024-11-01

**Soundness:** 2
**Presentation:** 2
**Contribution:** 2
**Rating:** 3
**Confidence:** 4

**Summary:**

This paper proposes a Spatial-Temporal Graph Attention Network with hierarchical transformer (STGAT) architecture for predicting forex rate. This model addresses two major challenges in forex rate prediction: (1) capturing interdependencies among different currencies and (2) managing long-range temporal dependencies. STGAT combines spatial graph convolutions with a dual-view temporal transformer, incorporating a novel temporal linearity graph attention network (TLGAT) for efficiency. The model outperformed ten baseline models across three categories (regression-based, transformer-based, and GNN-based) in experiments with 17 currencies over 2,092 trading days.

**Strengths:**

S1: The combination of spatial graph convolutions and dual-view temporal transformers addresses the limitations of traditional forex models by capturing inter-currency dependencies and effectively handling long-range temporal dependencies for improved prediction accuracy.

S2: The implementation of a linear attention mechanism reduces computational complexity, enhancing the model’s suitability for large-scale forex data involving numerous currency pairs.

**Weaknesses:**

W1: There are minor typographical issues in the paper: "firstly" is repeated on lines 240 and 243; "EXPERIMENTAL SETTTINGS" on line 313 should be "EXPERIMENTAL SETTINGS"; and "Traing setup" on line 321 should be corrected to "Training setup."

W2: The description of the model's input and output in Figure 2 is unclear.

W3: The paper's writing has considerable room for improvement, as the definitions of symbols in the methodology section are rather ambiguous.

W4:While you claim that your model effectively captures long-term dependencies, the experimental setup lacks clarification on the number of time steps used as the model's input.
W5: In Section 3.3, where you mention that "as a result of the Brexit event, there will be an increase in the volatility of the Pound, especially in the face of uncertain events," it would strengthen your analysis to include specific examples. For instance, in 2023, when the U.S. Federal Reserve raised interest rates, the resulting increase in the dollar’s exchange rate impacted forex rates globally, leading to a depreciation in other currencies, especially those with close economic ties to the U.S. Similarly, other geopolitical or economic shifts, such as fluctuations in oil prices or trade policies, could be cited to illustrate how such events amplify volatility across forex rates, including the Pound.

**Questions:**

Q1: For W2, could you provide a more detailed description of each module’s inputs and outputs?
Q2: For W4, could you analyze the model's performance on both long-input and short-input scenarios? Additionally, please specify how many future time steps your model is predicting.
Q3: For W5, could you include an analysis of these financial events? For example, identify specific phases where predictions were less accurate, possibly due to complex financial conditions coinciding with particular events.

---

### Official Review · Reviewer_M6hf · 2024-11-03

**Soundness:** 2
**Presentation:** 3
**Contribution:** 2
**Rating:** 3
**Confidence:** 3

**Summary:**

This paper introduces a novel Spatial-Temporal Graph Attention Network with Hierarchical Transformer (STGAT).

**Strengths:**

1.	The model architecture is novel, introducing a spatial-temporal graph attention network and hierarchical transformer for Forex rate forecasting.
2.	The presentation of methods is clear, and the performance metrics, MAE and RMSE, are reasonable.
3.	The research contributes to financial time series analysis and can be generalized to other domains like stock prediction.

**Weaknesses:**

1.	The combination of spatial graph convolutions with transformers has been explored in other domains, and the paper does not adequately highlight the unique challenges of Forex forecasting that the proposed model addresses.
2.	In the experiments part, line 316, the dataset of 17 currencies against the Chinese Yuan is limited. It should be discussed whether the chosen data can generalize to other market behaviors.
3.	The performance of graph construction from line 230 with k-means should be examined to determine if the final graph accurately represents the real situation, you should give analysis in the experiment part.

**Questions:**

1.	Could you provide some cases and analysis for graph construction?
2.	Were experiments conducted on currencies other than the Chinese Yuan?
3.	Given that research on Forex rate prediction is limited, have you considered using methods from stock prediction as a baseline, since they are similar?

---

### Meta-Review · Area_Chair_1NLX · 2024-12-17

**Metareview:**

This paper proposes a model by integrating a spatial-temporal graph attention network with a hierarchical transformer and applies it to the forecasting of currency exchange rates.

Major strengths:
- Forex forecasting is an important application in financial time series forecasting and the proposed model seems to address some limitations of existing models for forex forecasting.
- The proposed method has potential to be generalized to other financial time series forecasting applications, though they are not included in this work.

Major weaknesses:
- Though not for financial time series forecasting applications, the combination of spatial-temporal graph convolutional networks with transformers has been studied in other domains. The paper does not highlight the unique characteristics and challenges of forex forecasting as compared to the other applications studied.
- The experiments can be strengthened, which includes incorporating appropriate data normalization, applying hyperparameter tuning, expanding the dataset to include more currencies, and analyzing if the results are in line with the real situations.

The paper in its current form is not up to the acceptance standard of ICLR. The authors are encouraged to improve their paper for future submission by considering the comments and suggestions of the reviewers.

**Additional Comments On Reviewer Discussion:**

The authors did not respond to the reviews.

---

### Decision · Program_Chairs · 2025-01-22

Reject